



# ICESat-2 surface elevation assessment with kinematic GPS and static GNSS near the ice divide in Greenland

Derek J. Pickell[1], Robert L. Hawley[1], Denis Felikson[2], and Jamie C. Good[1]

[1]Department of Earth Sciences, Dartmouth College, Hanover, New Hampshire, USA
[2]NASA Goddard Space Flight Center, Greenbelt, Maryland, USA

**Correspondence:** Derek J. Pickell, derek.gr@dartmouth.edu

**Abstract.** Since 2007, researchers have conducted monthly or bi-monthly kinematic GPS surveys along a 15-km transect near Summit Station, Greenland, providing ice surface elevation data with high relative accuracy ($\pm$0.8 cm) and high precision ($\pm$0.8 cm). We use these surveys to assess the long-term stability of ICESat-2 surface height measurements, revealing a sub-1.0 cm bias and sub-6.0 cm precision relative to ICESat-2 data, with no significant temporal trend in performance. While reliable, these surveys are resource-intensive. We introduce an alternative, novel validation method using autonomous GNSS stations with interferometric reflectometry (GNSS-IR) to measure surface elevation concurrent with ICESat-2 overflights. This method agrees well with kinematic GPS (-0.2 $\pm$ 5.0 cm) and is sensitive to active accumulation and surface roughness, offering additional environmental context. The ICESat-2 measurements are biased by -0.9 $\pm$ 3.8 cm compared to these autonomous stations. Together, these results demonstrate the importance of sustained, high-accuracy GNSS for building a long-term elevation benchmark record in Greenland, while also establishing GNSS-IR as a scalable alternative in support of current and future altimetry missions.

## 1 Introduction

Precise and accurate satellite altimetry is required to measure the Greenland Ice Sheet's accelerating contribution to sea level rise (e.g., Sørensen et al., 2011; Khan et al., 2022; Smith et al., 2023c). Early measurements of Greenland's elevation changes relied on airborne radar campaigns in the 1980s and 1990s, which had limited accuracy and coverage, particularly in steep terrain. The launch of the Ice, Cloud, and land Elevation Satellite (ICESat) in 2003 significantly improved repeat coverage with laser altimetry, enhancing the understanding of surface mass balance of peripheral glaciers (e.g., Howat et al., 2008), the margins and interior of the Greenland and Antarctic ice sheets (e.g., Zwally et al., 2011), and ice shelves (e.g., Brunt et al., 2017), while achieving centimeter-level precision and accuracy (Schutz et al., 2005; Abdalati et al., 2010; Siegfried et al., 2011).

ICESat-2 improves on ICESat with continuous measurements on a 91-day repeat cycle, a six-beam array for slope detection and correction, and a smaller 17 m footprint for each beam (Markus et al., 2017; Magruder et al., 2021). Several efforts have assessed ICESat-2 performance in the cryosphere, primarily through ground campaigns in Antarctica. A 300 km traverse at 88° S, conducted annually from 2018 to 2020, intersects ∼275 ICESat-2 reference ground tracks (RGTs) due to orbital convergence





at this latitude (Brunt et al., 2019, 2021). Low accumulation in this region allows ICESat-2 surface elevation estimates to be compared directly to ground traverse data, regardless of the timing of recent satellite overpasses. These studies have evaluated all six ICESat-2 beams, building on validation approaches from the original ICESat mission and airborne lidar campaigns (e.g., Fricker et al., 2005; Schröder et al., 2017; Brunt et al., 2017).

At Summit, Greenland (72.573° N 38.470° W, 3253 m), a ground team has conducted monthly or bi-monthly kinematic GPS ground traverses since 2007 on a defined survey transect, making it unparalleled as a consistent, temporally long record of surface elevation change (Figure 1a). These surveys provide a coincident record for the entire ongoing seven year duration of the ICESat-2 mission. By contributing a complementary geographic setting to Antarctica, these surface elevation measurements expand on previous validation efforts while reducing uncertainties from temporal offsets between overflights and ground surveys. Unlike the arid East Antarctic Plateau, Summit experiences a significant accumulation rate, offering a realistic yet controlled environment to ensure ICESat-2 accurately captures ice sheet surface changes.

From 2022 to 2025, a network of low-power, low-cost, static on-ice Global Navigation Satellite System (GNSS) receiver stations, called OGREs (Open GNSS Research Equipment) operated along the ICESat-2 GPS traverse and at other regional ICESat-2 overflight locations (Fig. 1a) (Pickell and Hawley, 2024b). With Summit accumulating approximately 0.6–0.8 m y$^{-1}$ (Pickell et al., 2025; Dibb and Fahnestock, 2004), these autonomous stations further reduce accumulation-related uncertainties by recording surface elevations temporally coincident with ICESat-2 overflight times. Surface elevation is derived from each OGRE geolocated antenna position using GNSS interferometric reflectometry (GNSS-IR), which determines the antenna height above the snow/ice surface using ground-reflected GNSS signals (Larson et al., 2009). This technique is widely applied, including for tidal measurements and snow depth estimates (e.g., Larson et al., 2013; Larson and Nievinski, 2013). Pickell et al. (2025) also show that this technique is sensitive to surface roughness and sub daily accumulation in this region, which may further contextualize ICESat-2 surface height and height uncertainty estimates.

Here, we evaluate the data quality of kinematic GPS ground traverses (2018–2025) and static GNSS stations (2022–2025) to assess their accuracy and precision for comparison with precise satellite altimetry. We then compare these measurements with the full record of coincident ICESat-2 surface elevation data, analyzing the temporal stability of the ATLAS instrument over its lifespan and across all seasons. Finally, we analyze GNSS-IR sensitivity to surface and near-surface environmental conditions during ICESat-2 overflights, and whether these environmental conditions such as active accumulation, blowing snow, or clouds affect ICESat-2 data.

## 2 Data

### 2.1 ICESat-2 ATL06

For ICESat-2 traverse data comparisons, we select data from RGT 749 and RGT 879, along with nine additional RGTs for OGRE comparisons (Fig. 1a). Our analysis uses the land-ice elevation product ATL06 (version 6), which determines surface elevation estimates by selectively fitting and filtering 40 m windows of along-track geolocated photons every 20 meters (Smith et al., 2019). ATL06 serves as the foundation for many higher-level ice surface change datasets.





**Figure 1.** (a) Map of the ICESat-2 traverse route, Open GNSS Research Equipment (OGRE) GNSS-IR stations, and ICESat-2 reference ground tracks for laser beams 2L and 2R. The inset highlights the northern traverse end where ascending and descending ICESat-2 tracks intersect at OGRE station 879N. (b) Plan view of example OGRE station position and sensing footprint relative to the 2L and 2R ICESat-2 beams. The inset depicts an example azimuthal distribution of GNSS-IR antenna height estimates $H_r$ over a 24 hr period. (c) Survey setup: the red sled, carrying a Trimble R7 receiver and roof-mounted antenna, is towed by a snowmobile. Sled sinkage ($Z_{track}$) is measured twice each survey to correct GPS heights to the snow surface. To the right, the static OGRE setup features an antenna planted on a pole in the firn, whereby the antenna height $H_r$ is measured using GNSS-IR. Surface elevations on this map are derived from the MEaSUREs Greenland Ice Mapping Project 2 (Howat et al., 2014).





We subset the data to include only elevation estimates from Spot 3 and Spot 4, the middle beam pair that intersects the
ground traverse route and OGRE sensing footprints (Fig. 1a, b). Spot 3 is approximately four times stronger than Spot 4 due to
ATLAS instrument design constraints (Markus et al., 2017). Every few months, the strong and weak beams switch orientation
as the satellite performs a 180° yaw maneuver, thus switching Spot 3 and Spot 4 correspondence with the 2L and 2R tracks.

## 2.2 Kinematic survey GPS data

We collected GPS data at 1 Hz using a Trimble R7, with surveyors driving at approximately 5 m s$^{-1}$ along the $\sim$25 km
route (Fig. 1a, c), which intersects ICESat-2 RGT 879 and RGT 749 at multiple locations. At this speed, each GPS solution
corresponds to a $\sim$5 m footprint. To minimize the impact of accumulation-driven surface changes, surveys are conducted within
five days of the ICESat-2 overflight.

We processed GPS data using the Canadian Spatial Reference System Precise Point Positioning (CSRS-PPP) platform
(Natural Resources Canada (NRCan), 2025). Processing parameters include final IGS orbit solutions, a 7.5° elevation angle
cutoff for multipath mitigation, and corrections for solid Earth and polar tides. CSRS-PPP only fixes integer phase ambiguities
with GPS signals. Solutions are referenced to the GRS80 ellipsoid in the corresponding ITRF frame at the epoch of data
collection, with IGS antenna phase center offset corrections applied for the Zephyr antenna, ensuring final solutions correspond
to the antenna reference point (ARP). ICESat-2 land surface heights are similarly corrected for solid Earth and polar tides but
are referenced to the ITRF2014 frame and WGS84 ellipsoid (Neumann et al., 2019), and we transform CSRS-PPP solutions to
the ITRF2014 frame referenced to the WGS84 ellipsoid.

To derive surface elevation, the antenna mount height (1.797 m) is subtracted from the ARP. Track depth $Z_{track}$, the snow
surface depression due to vehicle weight, is measured at the start and end of each survey, with the mean value added to
the corrected elevation estimate. During the summer of 2024, we also mounted a downward-pointing laser rangefinder with
millimeter-level precision on the survey sled sidewall to monitor sled track depth, providing an independent estimate of sled
surface penetration variability.

## 80 2.3 OGRE GNSS and interferometric reflectometry data

Each OGRE station features an L1/L2 frequency GNSS antenna mounted on a pole above the snow (Fig. 1c). These stations
span from the west of Summit Station, where the maximum estimated slope is $\sim$0.09° to near the ice divide ($\sim$0.00°), where
an additional station (Station 1260) was deployed in 2023. During the summer of 2024, all but the three closest stations to the
ICESat-2 traverse route were decommissioned.

The OGREs collected multi-constellation data at 1 Hz during 24-hour periods coinciding with the 12 hours prior and the
12 hours after an ICESat-2 satellite overflight at the OGRE's location. These data were processed statically using CSRS-PPP
with the same parameters as the kinematic dataset. The westernmost OGREs experience significant horizontal displacement at
approximately 2 m y$^{-1}$ (5.5 mm day$^{-1}$), which introduces additional uncertainty into the PPP solution. Nevertheless, we treat
the station as static and accept the positional error from unmodeled motion. Meanwhile, Station 1260 logged data for only 3
90  hours daily, which allowed it to run daily throughout the winter. The shorter logging period increases uncertainty in the vertical



component of the PPP solutions, but we included this station in our analysis due to its unique configuration and geographic location.

To convert from the antenna height ($H_r$) to the snow-air interface, we use the GNSS-IR technique and follow the methods of Pickell et al. (2025) and Larson (2024). This produces hundreds of $H_r$ estimates distributed circularly about the instrument (Fig. 1b). We then take the mean of all $H_r$ estimates ($\bar{H}_r$) over a measurement period, and subtract it from the statically-determined vertical elevation of the antenna phase center, $Z_{ppp}$:

$$OGRE_{surface} = Z_{ppp} - \bar{H}_r. \tag{1}$$

Phase center corrections to the ARP are not applied to the geodetic elevation in the CSRS-PPP platform. However, uncertainty from this omission cancels when $\bar{H}_r$ is subtracted from the static elevation, as $\bar{H}_r$ estimates are relative to the antenna phase center.

We also analyze individual $H_r$ estimates over each measurement period to determine if there is active accumulation or erosion taking place on the surface; a statistically significant temporal trend in which $H_r$ estimates change by more than 1.0 cm during a 24 hour window is flagged as a potential period of significant wind redistribution, precipitation, or hoar frost events.

## 3 Data quality assessment and comparison methods

### 3.1 Kinematic GPS data quality assessment

Following the methods in Siegfried et al. (2011), we assess the quality of the kinematic survey data by analyzing the scatter of positions during periods when the survey sled is stationary for 7 seconds or more. The median standard deviation of each of these "pseudo-static points" (PSPs) across all surveys is calculated to be 0.8 cm (n = 9353), representing the precision of each individual GPS elevation estimate. This compares similarly with a prior estimate of precision of 0.9—1.8 cm (n = 108) at Summit with the Trimble R7 (Siegfried et al., 2011).

To assess the accuracy of the data, we take advantage of unique surveys where the traverse is driven twice, allowing us to compare repeat survey data minimally affected by surface changes. For each position in the first survey, we find the nearest corresponding point from the second survey within a 5 m radius and calculate the elevation residuals. This method introduces some uncertainty due to small-scale spatial variability but avoids uncertainty from interpolation to a common location. The median residual across all points from all repeat surveys is 0.8 cm (n = 78,025), representing the relative accuracy of each GPS-derived elevation estimate. Siegfried et al. (2011) estimated a relative accuracy of 0.9 cm (n = 1059) using a similar method, and Brunt et al. (2017) estimated a relative accuracy of -0.9 to 2.6 cm (n = 1067), which closely agree despite slight differences in data processing and comparison methods.





## 3.2 Kinematic GPS data comparison with ICESat-2 ATL06

To assess the bias and precision of ICESat-2 ATL06 estimates, we first search for any GPS ground survey within 4 days before or after the overflight date. ICESat-2 ATL06 data are filtered to include only those with a quality flag of 0 (`ATL06_quality_summary`), indicating no detected quality issues with the segment. For this flag to be zero, certain photon density criteria must be met, along with low errors in surface heights: this flag is designed to conservatively ensure bad data is filtered, sometimes at the expense of flagging good data (Smith et al., 2023b). Given the study area is low-sloping and crevasse-free, we apply further constraints on estimates of the ATL06 spread by decreasing the allowable height and geolocation errors for each ATL06 elevation: `h_li_sigma` <0.1 m, `geo_h_sigma` <1 m. These additional filters reduce the overall number of comparisons from 1823 to 1747. The nearest GPS point within a 20 m radius of the ATL06 point location is identified and used to calculate the bias of the ICESat-2 ATL06 elevation estimate.

To assess the long-term temporal stability of ICESat-2 surface elevation biases relative to the ground-based GPS measurements, we then fit a linear model to the residuals for each beam spot. We evaluated the significance of any temporal trend based on the slope of the fitted model. Additionally, to test for systematic differences between ascending and descending orbits, we performed a two-sample t-test comparing the distributions of elevation biases for each orbit direction. We assumed normality for both sample distributions in this test.

## 3.3 OGRE data quality assessment

The OGRE-derived surface elevation estimate, $OGRE_{surface}$ consists of two components: the PPP-derived elevations, and the GNSS-IR-derived correction of that elevation to the surface (Eq. 1). Due to the unique geometry of the $H_r$ reflection locations (Fig. 1b), it is challenging to analytically quantify the uncertainty of the mean elevation estimate for a 24 hour period, which is comprised of hundreds of $H_r$ estimates in different azimuthal directions throughout the period. We estimate uncertainties based on the $H_r$ mean standard error, the surface roughness estimate in the sampling domain, and a centimeter-level uncertainty for the PPP vertical solution (Eq. 2).

$$
\sigma_{mean,total} = \sqrt{\left(\frac{\sigma_{reflection}}{\sqrt{N}}\right)^2 + \left(\frac{\sigma_{spatial}}{\sqrt{N}}\right)^2 + (\sigma_{elevation})^2}. \tag{2}
$$

While PPP solutions report uncertainties at mm-level, past studies in the cryosphere confirm that this is often an underestimate (Khan et al., 2008). We chose an estimate of vertical uncertainty, $\sigma_{elevation}$, of 1.2 cm, which was determined from the same instrument and antenna combination as this study using a crossover analysis in Pickell and Hawley (2024b). $\sigma_{reflection}$ is the interferometric reflectometry precision, taken as 3.0 cm (Pickell et al., 2025), and $\sigma_{spatial}$ ranges from 2 - 9 cm based on estimates from Pickell et al. (2025). If we assume maximum values in Eq. (2), including a 9 cm spatial variability in surface elevation within the instrument footprint, and 200 reflectometry measurements in the 24 hr period, our estimated overall precision of the combined PPP - GNSS-IR surface elevation measurement is approximately 1.4 cm.





Past studies have shown GNSS-IR-derived surface elevations are biased (Pickell et al., 2025; Siegfried et al., 2017), and we assess this by comparing surface elevations to the kinematic GPS data when the kinematic survey passes within 20 m of the OGRE; this assessment yields a bias of -0.2 ± 5.0 cm (n = 72). We also extend the analysis from Pickell et al. (2025) which robustly compares GNSS-IR estimates to a network of manually measured stakes distributed within the sensing footprint of one of the OGREs between 2022 and 2025. We find a bias of 2.1 ± 2.9 cm (n = 125) (Fig. A1). Consequently, we correct for

this bias for all OGRE surface elevation estimates, noting that this value may be specific to this particular instrument-antenna setup (u-blox ANN-MB) in the dry snow zone of Greenland.

### 3.4   Comparison method: ICESat-2 and OGRE

We follow the same method described above to compare ICESat-2 ATL06 data with OGRE data, including filtering for a quality flag of 0 and a temporal search of 4 days (the majority of temporal offsets between ICESat-2 overflights and OGRE data are

0 days). However, as spot 3 and 4 straddle the OGRE, we must define a larger search radius to find intersecting ICESat-2 ATL06 points at 60 meters, which is greater than the 45 m across-track spacing between ATL06 beam pairs, to account for variability in ICESat-2 beam pointing. We assume a linear surface slope between each beam pair, and assess the excursion of the OGRE elevation from the linear interpolation. This approach will contain additional uncertainties due to topography and surface roughness variability which are unaccounted for in this interpolation, but this method will still allow us to evaluate

cross-track consistency between the two laser beams and identify potential beam biases.

   To assess how surface conditions influence ICESat-2 elevation retrievals, we used OGRE station data to flag days with active surface height change. We defined an active surface event as any 24-hour period with a detected surface height change greater than 1.0 cm. These events may reflect wind redistribution or snowfall, both of which can affect ICESat-2 return quality. Then, we identified corresponding ICESat-2 overpasses affected by blowing snow or cloud cover using the **bsnow_conf** and

**cloud_flg_asr**, retaining only those classified with at least medium confidence. If the processes driving surface height change on the surface indeed affect the optical return of the laser altimeter, then these events should be reflected in corresponding ICESat-2 snow or cloud flags.

   To further examine these potential environmental impacts on altimetry precision, we compared height residual distributions between days with and without blowing snow or cloud flags, and between days with elevated surface roughness (from OGRE-

derived $H_r$) and those without. We also tested for a correlation between increased residuals and increased surface roughness estimates, in addition to any correlation between ATL06 reported height uncertainties and OGRE-derived surface roughness.

## 4   Results

### 4.1   ICESat-2 - kinematic traverse comparison

Between 2018 and 2024, the ground traverse team successfully surveyed over 30 overflights, and 1747 comparisons were

made individually between ATL06 points and ground-based measurements (Fig. 2). Aggregated statistics of biases relative to





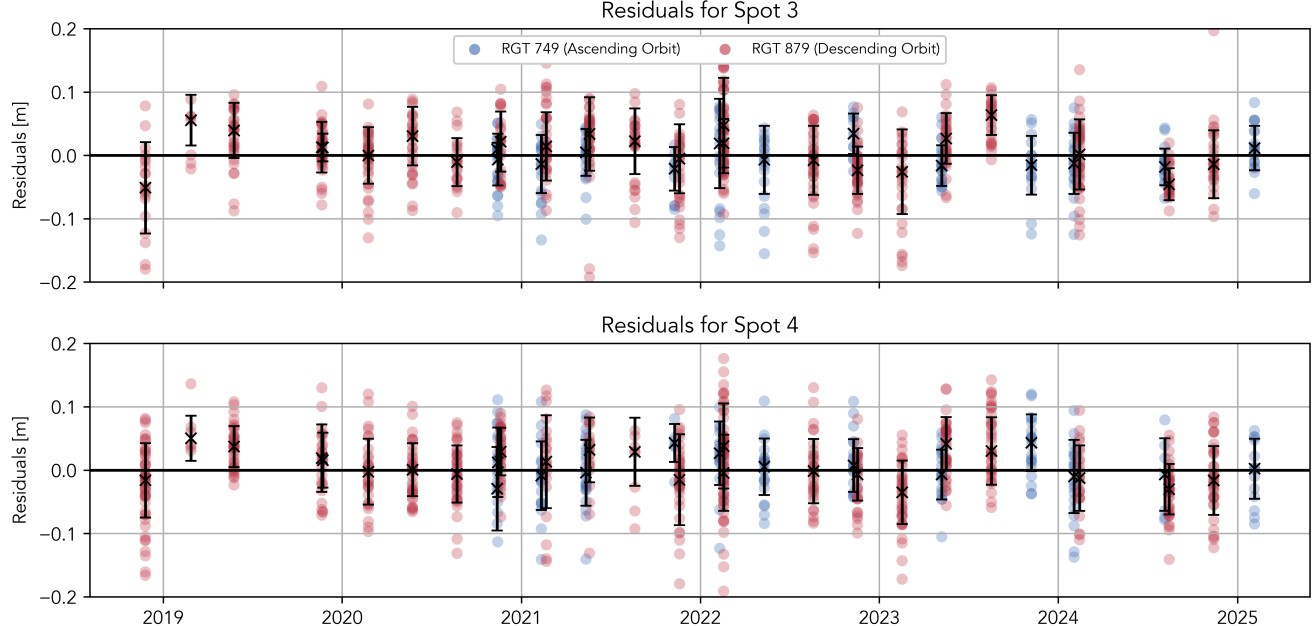

**Figure 2.** Spot 3 and Spot 4 residuals relative to kinematic GPS data. Residuals derived from descending overflight comparisons are in red, and residuals derived from ascending comparisons are in blue. We also show the median residual between the GPS survey and ICESat-2 ATL06 heights for each overflight in black, with the vertical bars indicating uncertainties expressed as the standard deviation of the residuals from that day.

the GPS over all overflights are presented in Table 1, alongside results from previous validation efforts in Antarctica. The 1-$\sigma$ standard deviations are taken to be the ICESat-2 precision relative to the GPS data.

To assess the long-term stability of the biases, we applied a linear fit to the ATL06-GPS residuals across the entire comparison period. No statistically significant slope was detected, indicating no measurable trend in bias over time. We also found no

significant differences between ascending and descending orbits (p > 0.05), suggesting orbit direction does not affect the bias.

**Table 1.** Comparison of ICESat-2 Elevation Biases from Different Validation Datasets

| | 88° S 2019-2020 Brunt et al. (2021) 20 m search | Summit ICESat Traverse (This Study - 20 m search radius) | Summit OGREs (This Study - 60 m search radius) |
|---|---|---|---|
| **Spot 3** | $2.2 \pm 6.0$ cm | $0.3 \pm 5.5$ cm | |
| | ($n = 1,889$) | ($n = 847$; 35 overflights) | -0.9 $\pm$ 3.8 cm |
| **Spot 4** | $1.6 \pm 6.6$ cm | $0.5 \pm 5.7$ cm | ($n = 192$ comparisons; 76 overflights) |
| | ($n = 1,835$) | ($n = 900$; 36 overflights) | |



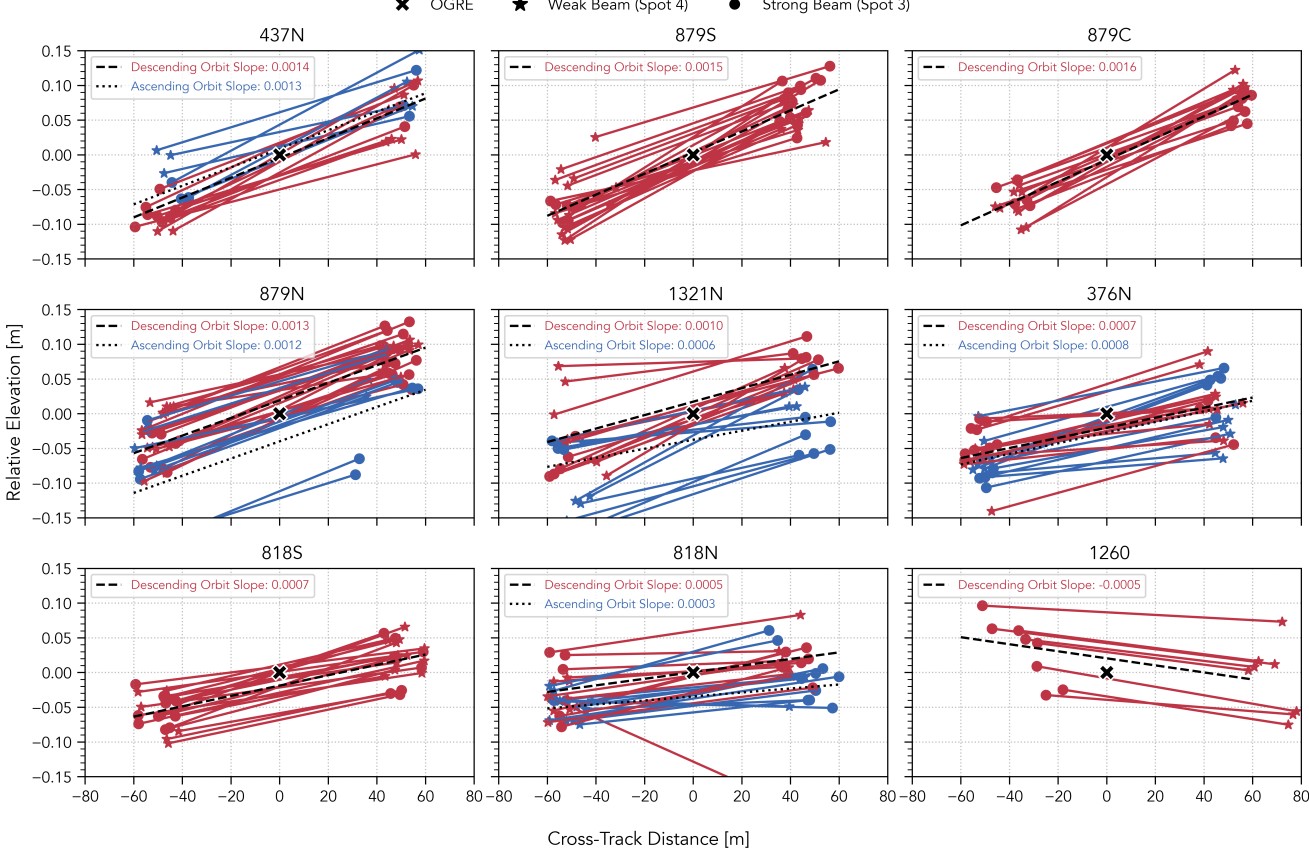

**Figure 3.** ICESat-2 beam pair elevations relative to OGRE-derived surface elevations. Any beam pair found within the 60 m search radius of the OGRE is plotted and colored by the orbit direction. The end-cap markers are stars for the weak beam spot and circles for the strong beam, which indicate the satellite's yaw configuration.

## 4.2 ICESat-2 - OGRE comparison

While the ICESat-2 ground traverse can intersect at most 10 overflights in a year (tracks 879 or 749 are repeated every 91 days), the OGRE network covers 11 unique RGTs, allowing for dozens of comparisons in a year. Over the study period, 76 overflights took place, from which 192 Spot 3/Spot 4 beam pairs fall within the 60 m search radius of the OGREs, producing a median bias of -0.9 ± 3.8 cm (n = 192) (Table 1, Fig. 3). Over this shorter temporal period, a larger difference in biases between ascending and descending orbits is observed; surface estimates from ascending tracks are -2.3 ± 4.3 cm (n = 51) compared to -0.5 ± 3.3 cm (n = 141) for the descending tracks. No statistical differences are observed in the slope estimates when the strong and weak beams switch sides relative to the OGRE, which would arise if one beam was significantly more biased than the other.



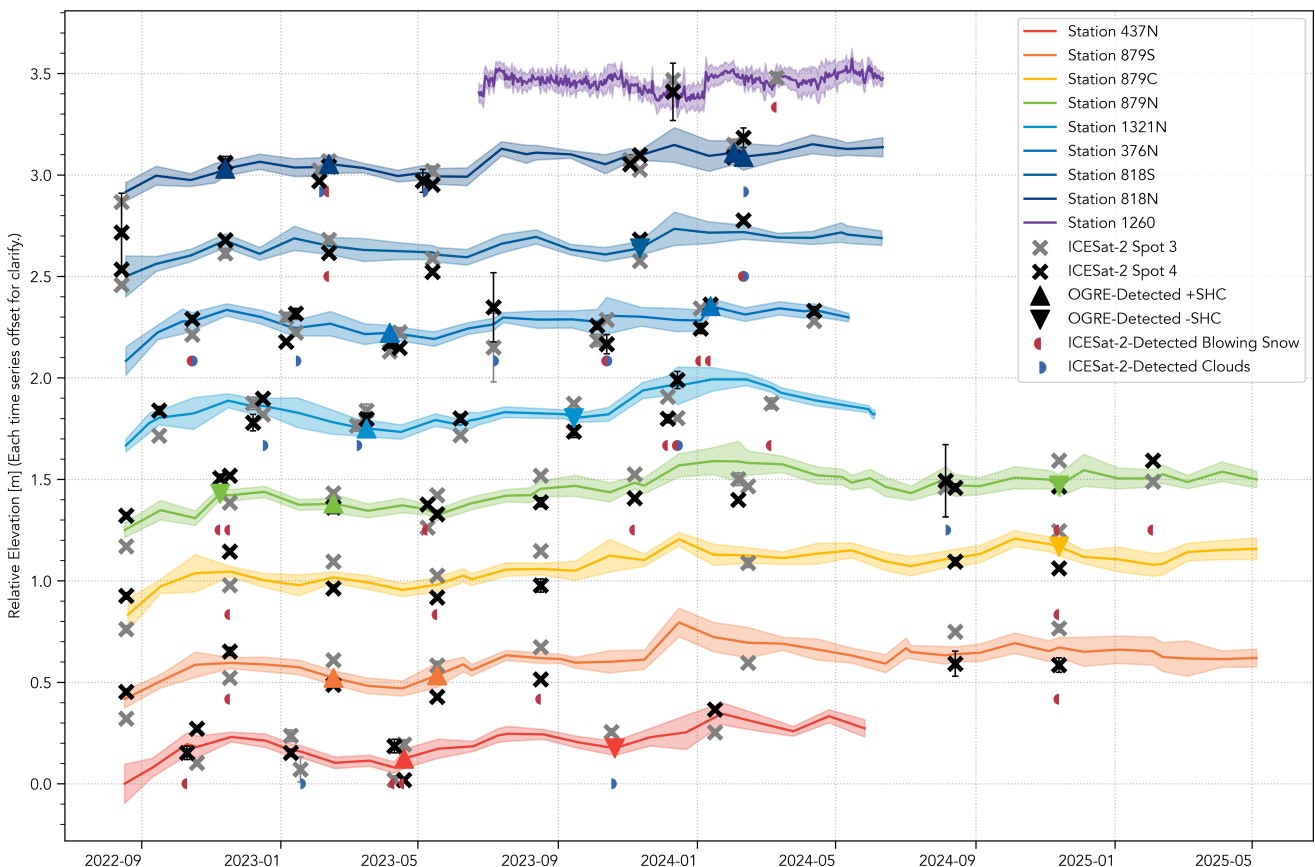

**Figure 4.** Time series of OGRE surface elevations compared to adjacent ATL06 surface elevations. Each OGRE time series is vertically offset by +0.4 m for clarify and annotated to indicate periods of active positive or negative surface height change (SHC) corresponding to ICESat-2 overflights. The 1-$\sigma$ shaded regions surrounding each OGRE time series represent the spread in $H_r$ estimates, which are influenced in part by surface roughness. Elevations and uncertainties are derived from monthly or bi-monthly estimates (or daily estimates for Station 1260), but are connected with lines for visual continuity. ICESat-2 elevations falling within each hemispherical footprint of each OGRE station are averaged together, and the error bars are taken to be the spread in these data, or in the case of only a single elevation, the ATL06 estimated error of that point. Each overpass is annotated with a marker when blowing snow or clouds were detected.

For the assessment of potential environmental influences on ICESat-2 data quality, we first identified 14 days in which OGRE-detected surface height changes were greater than 1.0 cm. Meanwhile, blowing snow or cloud cover was detected by ICESat-2 flags on 23 days, but only 4 of these coincided with days when OGREs detected surface height change. Furthermore, when comparing ICESat-2 residuals between flagged and unflagged days (i.e., cloudy or blowing snow vs. clear-sky), we found no significant differences in the elevation biases. Similarly, we observed no relationship between OGRE-derived surface roughness and ATL06 height uncertainty (`h_li_sigma`), the propagated measurement of error for ATL06 measurements.





## 5 Discussion

Temporal biases or long-term trends in altimetry performance can affect mass balance studies (Siegfried et al., 2011), yet the absence of a clear temporal trend in our comparison data shows ICESat-2 altimetry consistency beyond its original three-year mission. Meanwhile, the correlation between each overflight median bias for the two spots is r = 0.64 (p <0.01), indicating that the biases are influenced by the same physical or instrumental effects. Other sources of variability in biases may stem from time-dependent volumetric/subsurface scattering of the beams, though in interior regions, this effect on ATL06 estimates is likely under a centimeter (Smith et al., 2019, 2025). Rain events, thick hoar frost, and potential seasonal differences between winter and summer snow pack are not captured by the ATL06 algorithm. Furthermore, atmospheric forward scattering increases in the presence of clouds or blowing snow, but is not corrected during ATL06 processing (Smith et al., 2019) and our results do not indicate this effect is detectable.

The temporal offset between the ICESat-2 overflight and the ground survey also introduces uncertainties as the surface evolves in the intervening time. However, the reduction of the temporal search to a $\pm$ 12 hour period for the kinematic data does not significantly change our results. The median bias and precision becomes -0.1 $\pm$ 5.9 cm for Spot 3 and 0.2 $\pm$ 5.7 cm for Spot 4 (compared to 0.3 cm and 0.5 cm originally), while the overall number of residuals is reduced by nearly 50%. We also investigate the spatial contribution to the uncertainty by increasing the search radius to 40 m, yielding biases of 0.0 $\pm$ 6.0 cm (n = 1719) and 0.2 $\pm$ 6.3 cm (n = 1782) for Spot 3 and 4, respectively. The small lowering of the ICESat-2 ATL06 surface relative to the kinematic traverse is accompanied by a larger spread in the precision estimate. More GPS points with the larger radius may better capture a spatially representative and similar surface to the ATL06 footprint, but also increase the uncertainty due to the increased spatial search area.

As most static OGRE data temporally straddle the satellite overflight, the temporal source of uncertainty is largely eliminated. However, the configuration of the OGRE stations far from the ATL06 centroids mean the precision estimate is more affected by the search radius. For example, when the search radius is expanded to 100 m, the bias remains stable (-0.9 cm), but the precision estimate decreases to $\pm$ 4.5 cm (n = 577). With the OGRE data, we also observe a statistically significant difference in orbit-based biases between the ascending and descending tracks within the OGRE comparison time frame. Luthcke et al. (2005) and Neumann et al. (2019) both mention potential differences in orbit determination and related parameters that may introduce short-term biases in photon estimation and geolocation. However, these effects may average out over time, appearing random on longer timescales, consistent with the absence of a temporal trend in our kinematic survey results.

Herzfeld et al. (2021) show that ICESat-2 is capable of detecting blowing snow and optically thick clouds, and is likewise affected by the presence of these features. Surface change in interior Greenland is often linked to blowing snow and/or clouds (Pettersen et al., 2018), and therefore we assume OGRE-detected height changes are accompanied by blowing snow and/or optically thick clouds. When we segregate data in our analysis from ICESat-2-flagged blowing snow and/or cloudy days, or days in which the OGRE detects active surface change, results do not yield better bias or precision estimates. Likewise, when we test for the effect of solar elevation angle (day versus night), no differences in residuals are found. Pickell et al. (2025) show that GNSS-IR is sensitive to changes in surface roughness at centimeter to meter scale, which would also affect photon



returns and potentially bias ICESat-2 height estimates. This sensitivity to roughness is captured by the uncertainty estimates for each static station in Fig. 4. However, we detect no significant correlation between ICESat-2 biases and surface roughness over this period. The lack of detectable response to environmental conditions may suggest that the ATL03 and ATL06 algorithms coupled with our outlier elimination successfully identify and filter data affected by these phenomena, and therefore do not impact our comparison results with the refined ATL06 product.

While considered baseline measurements, our effort to quantify the accuracy and precision of the GPS and GNSS measurements does not account for correlated errors such as those caused by tropospheric or ionospheric effects (Brunt et al., 2019). Another systematic effect may also stem from the variable track $Z_{track}$ depth throughout each survey. To investigate this, we examine the data from the downward-pointing survey sled laser over two traverses. The 1-$\sigma$ variability in track depths was ~2.0 cm. Since only two manual measurements are typically used to estimate the mean track depth for correction, this introduces a potential bias in the GPS-derived surface elevations due to under-sampling, with an estimated standard error of $\pm 1.41$ cm based on the laser-derived variability. This uncertainty should be propagated into any comparison with ICESat-2 surface elevations and may partially account for both the observed bias and the scatter in the residuals. OGRE surface estimates are free from this source of uncertainty. Finally, unique to the GNSS-IR method, the sensing footprint shrinks as the pole becomes buried, potentially increasing the sensitivity to localized topography or surface roughness. There may also be unresolved biases due to radio shadowing or seasonal effects that warrant further investigation (Appendix A).

## 6   Conclusions

The long-term campaign of ground-based GPS traverses in the interior region of Greenland shows high precision and relative accuracy, forming a robust dataset for ICESat-2 comparisons that cover the full ongoing mission lifespan. In comparing this ground-based data with ATL06 height estimates, biases are sub-1 cm and precision is better than 6.0 cm, consistent with past assessments conducted in Antarctica. Meanwhile, lower surface roughness and smaller temporal offsets between the ground surveys and overflights in Greenland may account for small improvements in results.

We also present an autonomous method of retrieving ground-based surface elevation estimates using GNSS interferometric reflectometry with a standard GNSS receiver, mounted on a mast in the snow. The unique capabilities of this setup include the ability to retrieve surface heights exactly aligned with ICESat-2 overflights, to eliminate the need to measure sled track depths, and to detect heightened surface roughness states and active surface height change. The resulting comparisons with this technique and ICESat-2 altimetry closely agree with the biases and precisions derived with the traverse data. Moreover, the high quality of the ICESat-2 surface elevation estimates led to no degradation in performance during ground-detected surface elevation change events coincident with ICESat-2 overflights.

The cost effectiveness and geographic flexibility of static stations enable more frequent and spatially distributed ICESat-2 comparisons, providing a scalable alternative method for current and future validation efforts requiring more autonomy and year-round reliability. By combining high-precision GPS traverses with static GNSS-IR observations, these findings reinforce



the high quality of ICESat-2 surface elevation data in interior Greenland and advance broader efforts to monitor cryosphere change with increased temporal and spatial resolution.

*Data availability.* The Ice, Cloud, and land Elevation Satellite-2 (ICESat-2) data used in this paper (ATLAS/ICESat-2 L3A Land Ice Height
ATL06, Version 6) are available from the National Snow and Ice Data Center (NSIDC) (Smith et al., 2023a). The OGRE data are available
at the Arctic Data Center (Pickell and Hawley, 2024a). The GPS traverse data are available at conus.summitcamp.org/mirror/summit/ftp/
science/ICESat/ICESat_data/.

**Appendix A:  Analysis of absolute bias of GNSS-IR technique for surface altimetry validation**

We extend the GNSS-IR bias assessment work of Pickell et al. (2025) by using a network of 121 snow stakes surrounding a
statically-placed OGRE near Summit Station. Pickell et al. (2025) found that changes in snow height detected from the OGRE
most closely corresponded to the inner 36 stakes that surrounded the OGRE, spaced 8 m apart in a grid pattern. These stakes
were referenced to the same datum as the OGRE antenna during the summer of 2024 to allow for direct comparisons between
the GNSS reflectometry-derived height above the surface and the mean stake height above the surface.

As the stakes and the OGRE were subsequently raised later in 2024, we repeated the automatic leveling of the stakes in
2025 to add an additional year of comparisons between the two datasets. We find the median bias between the two datasets
to be -2.1 $\pm$ 2.9 (n = 125), meaning that the OGRE reflectometry estimates are smaller than the stake height measurements.
Therefore, the OGRE surface, as measured from the antenna, is vertically higher than the automatically leveled stake surface
by approximately 2.1 cm.

We performed an ANOVA test (F=4.82, p=0.0033) showing seasonal variation in residuals, with the wintertime period
(December, January, February) having the largest residuals, and the summertime (June, July, August) having the smallest
residuals. This could be due to a number of factors, but it highlights the need to continue to study the sources of bias in GNSS
interferometric reflectometry. Snow buildup and pitting is most often observed in the wintertime in the stake forest and could
bias these poles, e.g.

*Author contributions.* DJP and RLH designed and executed the study and DJP wrote the manuscript, with feedback from all other authors.
DF provided ICESat-2 data analysis and feedback on the manuscript. JCG and RLH conceived of the downward-looking laser study.

*Competing interests.* The authors declare that they have no conflict of interest.





**Figure A1.** Time series of OGRE and stake-derived surface, coincident to the same area.

*Acknowledgements.* This project was funded by NASA grant 80NSSC21K1876. We thank Battelle ARO and the dedicated crew at Summit Station for the field logistics and support, and especially the science technicians who carried out the monthly traverses throughout the years.



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
