# Peer review of "ICESat-2 surface elevation assessment with kinematic GPS and static GNSS near the ice divide in Greenland"

_EGUsphere, 2025_

## Author Comment (AC1)

We thank the reviewer for these valuable comments that improve this manuscript. Please see our responses to each comment below in blue.

General comments

This is a great study. We need regular altimetry validation studies that use observations from kinematic GNSS, and this study which uses a unique dataset that samples surface height change in winter and in summer fills an observational gap.  I think the manuscript should be accepted with minor revisions. I also have mostly general comments that the authors can take or leave.

My main criticism is that I think the manuscript could go into more detail about the surface elevation trend that you observe at Summit. Do we see an increase or decrease in mean surface height over the 2-year period you have observations? The density of your observations in space and time provide a unique constraint that I think could be used to describe in more detail the subtle surface elevation change signals that are propagating into the interior from more well documented change from the margins. Could you add in the conclusion or the discussion a short subsection that describes these elevation changes and speaks to the significance of this observation in the context of Greenland/Summit accumulation derived from reanalysis and your observations? Consider adding a section in the discussion:

Interior surface-height change detected by the OGRE network

Beyond bias characterization, the OGRE time series document a net rise in the interior surface of the Greenland ice sheet during 2022-2024..”

We grappled with whether to include information about elevation change in this manuscript as studying not only absolute biases but also the ability of ICESat-2 to correctly capture elevation trends through time is important. Furthermore, understanding the processes that contribute to elevation change in interior Greenland are important for many glaciological and climate applications.

In fact, we have a manuscript in preparation that examines these points on elevation change. We feel this belongs in a separate manuscript because the processes that contribute to elevation change are complex. We must take into account firn thickening/thinning, ice thickening/thinning, in addition to surface processes such as accumulation, thus requiring multiple additional datasets and analyses that we feel would distract from the scope of this manuscript.

It might be outside the scope of this study, but in the future, could you consider using dual frequency GNSS from ground-based radar surveys and UNAVCO kinematic GNSS data to increase the number of GNSS/IS2 crossover points? Much of this data exists and is fully processed on UNVACO/CReSIS servers already, and though most of this data was collected without monitoring sinking from sleds, the sled design/geometry is well constrained and photos from different seasons could be used  to calculate the sinking term and augment the year-round surveying described here for summer months over different

regions of Antarctica and Greenland for environments where the surface conditions are more rough near the coast.

This is a good suggestion: we have examined several existing on-ice GNSS studies, including the SMM3 UNAVCO/Earthscope station at Summit and the GLISN stations in south Greenland. These static stations can use reflectometry to derive the surface elevation. However, they were not deployed underneath ICESat-2 overpass lines and therefore required additional corrections for slope that make them too imprecise for this study. These sites, in addition to other kinematic surveys, would be very suitable for radar altimetry comparisons, given the larger footprint.

One other question I have is connected to the methodology and processing of the kinematic GNSS data and -IR data. I think with a base station at summit and the network in Greenland maintained by UNAVCO, it should be possible to process the kinematic GNSS using TRACK relative to base station solutions using software like GAMIT/GLOBK (or public solutions from repositories that host GNET and summit data). In the case of the OGRE, processing with GAMIT/GLOBK as part of a larger solution for Greenland may improve the relative surface height estimates and could be worth considering in the future. If you need help setting this up for future studies, we can connect after the review period (I don't think it affects any of your main conclusions here).

Any suggestions to make the data more useful in future studies are appreciated and we would happily explore this suggestion further.

The only other delicate suggestion I have is to perhaps make less strong claims about the originality of the autonomy of these systems. For instance, someone likely from unavco or pascal is raising these sensors to make it possible to do GNSS-IR over multiple seasons. This is a lot of work, and it's been done for quite a while at Summit, but also more remote sites. For instance, take the second paragraph of the conclusion:

"We also present an autonomous method of retrieving ground-based surface elevation estimates using GNSS interferometric reflectometry with a standard GNSS receiver, mounted on a mast in the snow."

This language makes it seem as though this is the first use of GNSS-IR for monitoring surface height change of ice sheets when most of the methods you've described are well established (and I think still require people to service the instruments?). I think these sentences could be modified to emphasize the novel application (surface tracking/altimetry validation) using an established method. I don't think This was intentional, and my suggestion is just to make this more clear.

We appreciate this suggestion. Indeed, several papers have made use of UNAVCO on-ice infrastructure for reflectometry (e.g., Larson, M MacFerrin, T Nylen 2020), who first suggest the applicability of their methods to altimetry validation. We will adjust our language accordingly in the introduction and elsewhere, while maintaining the novel aspect here: that these stations were designed and deployed specifically for this purpose (e.g., geolocated along ICESat-2 paths with appropriate antenna heights).

Below are minor suggested changes for style and content:

Figure 1: In panel B, it appears most of the reflections are coming from within this azimuth angle of 5 degrees, but that this zone doesn't overlap in this case with the icesat2 passes. Could similar figures be made for all sites to show how where the measurements you're making are relative to the the icesat-2 tracks.

Given the relative compactness of our network (~30 km east-west), the reflection zones and azimuth angles are mostly consistent from station to station. The fact that the elevation angle window emphasizes the surface area closer to the instrument than the ICESat-2 paths is perhaps our largest source of uncertainty with this method and we will adjust our language accordingly.

LN: 7-11 consider removing autonomy, and defining GNSS, GNSS-IR. Also choose GNSS or GPS (as I think you probably use solutions from all the satellites not just GPS?)

"while reliable, these surveys are resource-intensive. We introduce an alternative, novel validation method using Global Navigation Satellite System (GNSS) Interferometric Reflectometry…"

We will make this language more self-consistent and defined and consult with the editor about acronym definition here. GPS refers to the kinematic survey with the R7 receiver, which is old enough to only track GPS/GLONASS and here only tracks GPS.

L12-16 Consider quoting the bias and standard deviation directly. Revealing mean bias of *** +/- ** cm relative to ICESat-2.

These quantified details for each methodology/technique may be too detailed for these initial sentences but we will examine the appropriateness of this information here or elsewhere in the manuscript.

L23-31: "By contributing a complementary geographic setting to antarctica.."

This is a long and awkward sentence to me -consider revising, and reframing around Summit as a legacy validation site and link to icebridge and icesat-1.

Fixed.

LN 112-118 (Section 3.1): "The median standard deviation of each pseudo-static point..

This was a sentence that I felt could be shortened and combined with the second sentence:

The median standard deviation of the pseudo-static points was 0.8 cm (n=****), consistent with prior Summit estimates (0.9 -1.8 cm).

We will consider this suggestion.

LN 158- 166 (Section 3.4): "We follow the same method described above to compare ICESAT-2…"

Consider changing for clarity to: For ATL06 – ORGE, we apply the same filters (quality.= 0 ; …, but we use a 60 m search radius (beam-pair spacing 45 m) and a linear cross-track interpolation between Spots 3-4 at the OGRE latitude.

Changed.

LN 233- 239 (Section 5): Consider changing "When we segregate…" to "Separating overpasses flagged for blowing snow or clouds does not affect bias or precision."

Changed.

Appendix A: It looks like there was an idea that was not finished or completed. What was the example the authors intended to include here (e.g. …).

Fixed.

Minor comment (not necessarily for a single section):

Can you include a summary table of the parameters you used in the GNSSrefl code. It would also be great to include figures of the Fresnel zone for each receiver as this shows really explicitly the area that you sample from. Height solutions can be sensitive to fesnel zones and the threshold azimuth angle, and recording all this information in a table could help users who want to replicate this kind of study quickly (a lot of this is already well documented in the code). Also include information about which frequencies (likely both?) and constellations were used in the reflection solution.

We will ensure that the processing parameters are properly detailed for easy replication.

**Copy edits:**

LN 29: bi-monthly -> bimonthly, and consider rewriting for clarity

Changed to semimonthly.

LN 44: Consider citing Larson & Nievinski (2013), Seigfried et al., (2017), Hoffman et al., (2025), Trine et al., (2024), which have used -IR to measure accumulation. GNSS-IR is a powerful measurement technology that is still underused in the glaciological community. Citing these other studies can bring

awareness to this method and how it can be used in validation studies of surface height change and to understand near surface accumulation and firn densification.

Added these citations.

LN 190: delete temporal

Done.

Figure 4 caption: Clarify -> clarity

Done.

Section 5: snow pack -> snowpack.

Done.

Throughout: Use ~ throughout to approximate value.

Done.

Throughout: Overflight -> overpass .. I'm not sure what the community standard is here. Flight seemed odd to me, but I could be wrong.

Standardized to overpass.

Throughout: Sub daily -> subdaily

Done.

Throughout: Include space before units.

Done.

Throughout: 1-\sigma -> 1\sigma

Done.

Throughout: Consider abbreviating y^-1 to yr throughout and being consistent with abbreviation of s y, and day (d). I defer to the editor on this. I'm not sure what the best practices are for the cryosphere.

We will change to yr and double check with the editor.

---

## Author Comment (AC2)

We thank the reviewer for these valuable comments which will make this manuscript stronger. Please see our responses to each comment below in blue.

The manuscript entitled "ICESat-2 surface elevation assessment with kinematic GPS and static GNSS near the ice divide in Greenland" by Pickell and others details a new use for the Open GNSS Research Equipment (OGRE) GNSS-IR stations—validation of ICESat-2 surface height measurements. This study first uses data from repeat kinematic GPS surveys at Summit Station, Greenland to assess the stability of ICESat-2 surface height measurements, finding a <0.01 m bias and <0.06 m precision between kinematic GPS and ICESat-2 observations. The authors then present a new validation method using GNSS-IR interferometric reflectometry to measure surface elevation coincident with ICESat-2 passes over their study site near Summit Station. Ultimately finding a good agreement between GNSS-IR and kinematic GPS observations and a bias in ICESat-2 measurements of ~0.09 m. These results indicate GNSS-IR can be deployed instead of kinematic GPS transects which are significantly more resource intensive.

I found the manuscript to be very well written with a robust methodology and results section wherein the results were well supported by the presented data. The manuscript would benefit from an elaboration on some methodology (detailed below) and a reorganization and expansion of the discussion section. I have detailed these points below and include some additional minor comments. Overall this work is appropriate for the Cryosphere and I would therefore recommend minor revisions before publication.

Major comments

(1) I would like some clarification on the bias corrections applied to the dataset. Section 3.3, Lines ~155. Here a bias of 2.1+/-2.9 cm is reported, citing figure A1. I have a few questions regarding figure A1, but with the main concern that this bias is not temporally consistent. Is the bias applied across the full time series as stated? It appears from Figure A1b that there is indeed a large bias in 2022, a moderate bias in 2023, and somewhat in early 2025. Why not apply a variable bias correction? Or not at all as it is not convincing that a bias correction of 2.1+/-2.9 cm would improve your observations. Regarding Figure A1: Which OGRE is this data from? Where was the stake relative to this OGRE and the study area? Also was there a camera that allowed such frequent stake readings?

Assessing the temporal bias of GNSS-IR is an important and ongoing area of research in the reflectometry community. Traditionally, snow depth reflectometry studies have only measured the height of the GNSS mast, without much consideration to the true footprint of the reflectometry technique. We attempt to address the footprint issue by deploying an OGRE at the "Summit Station" marker in Figure 1, centered within a grid of 121 snow stakes spaced 8 m apart [See details in Pickell, Hawley & LeWinter 2025]. While these stakes are each measured to the nearest half centimeter, seasonal biases may still exist due to human error, seasonal pitting or mounding at the base of the stakes, etc. Thus, we must be cautious about the attribution of bias, and especially time-varying bias, to the reflectometry technique.

We strike a compromise here by applying the overall mean bias correction to the OGRE, but we will adjust our language and descriptions to make the answers to your questions clear and to point out that the most important comparison between the altimetry and OGREs is the low variability between the ICESat-2 and OGRE measurements, which is independent of the OGRE bias correction.

(2) Figures supporting methodology (either added to manuscript or supplement) L75-78: What is the range of track depth Z_track values due to vehicular weight depressing the snow at the beginning and end of your survey and the mean value used? L78-79: How did the laser range-finder measurements compare with the mean value of Z_track measured? Were there any systematic trends in this value? E.g., increasing depression along the survey track? A figure showing the Ztrack observations and the laser range finder observations would be beneficial.

Typical track depth values range from 0 to several centimeters depending on the survey and season/snow conditions, and in general correlate with corresponding qualitative data that detail recent snowfall (deeper tracks) or cold, clear conditions (shallower tracks, e.g.). We will update the language to clarify our methodology here:

"The manual track-depth measurements (typically two per survey) provide a sparse estimate of the mean track depression, while the laser range-finder provides nearly continuous measurements along the track. Using the laser-derived standard deviation (σ ≈ 2 cm), the standard error of the manual mean is estimated as ±1.41 cm, quantifying the uncertainty introduced by under-sampling."

Unfortunately, we don't have more laser data from other survey dates from within this study to construct a full statistical analysis across the entire study period, and in general must take the routine, manual measurements at the beginning and end of each survey to be representative of the survey-wide track depth. However, our results from the laser rangefinder surveys are encouraging in that there is no systematic drift in track depth (e.g., indicative of changing snow conditions) for those particular surveys, which confirms the assumption that snow conditions are largely uniform in this region (although with some degree of sastrugi noise).

(3) I found that the discussion would benefit from some slight reorganization to make it easier to follow for the reader, in particular by separating the discussion of temporal and spatial sources of bias which are slightly intertwined (mainly the details presented in the middle to the end of paragraphs). Overall, the discussion does have a good organization by moving from those errors to external factors (e.g., blowing snow/surface roughness) then (correlated errors). But again, the authors should be careful to group like-ideas together (e.g., a discussion of the sensing footprint on line 248-249: is this an uncorrelated error or should it be moved to the spatial or surface roughness paragraphs? If it is kept in its current position the similarity of the sources of errors in the data should be made more evident.

We will structure sections of the discussion to more clearly separate temporal and spatial sources of bias. Specifically, details related to the sensing footprint will be adjacent to the discussion of spatial or

surface roughness–related errors, so that similar sources of error are grouped together to improve readability.

(4) The Manuscript would also benefit from an expanded discussion of the implications, next steps, or synthesis of the work presented here. For example, the introduction and abstract mention that kinematic surveys are much more labor and resource intensive than the OGRE station deployment, since these results demonstrate OGRES are a useful tool to assess ICESat-2 surface elevations, what are the authors recommendations moving forward? I would be great to hear their thoughts for how future campaigns aiming to assess airborne/space-borne surface elevation measurements should proceed. These topics are particularly relevant given NASA's Snow4Flow program.

We agree that an expanded discussion of the implications and next steps would strengthen the manuscript. In the revised discussion section, we can highlight that OGRE deployments offer a lower-cost, logistically simpler alternative to kinematic GNSS surveys for evaluating ICESat-2 surface elevations. We note that while kinematic surveys remain valuable in specific settings, OGREs provide an efficient means to establish ground control across larger spatial and temporal scales, while reducing logistics. For example, radar altimetry validation could benefit from reflectometry given the reflectometry footprint is more agreeable with space-based radar, e.g. Or, with regards to Snow4Flow or NASA's NISAR mission, we can leverage the movement of static stations (which nominally is a challenge for co-located measurements) to validate ice flow and elevation change together.

Minor comments

Figure 1 (a): I would suggest the north arrow be positioned on the top of the figure as it seems out of place near the scale bar. The legend for ICESat Traverse Route is somewhat misleading

We will experiment with alternative placements of the legend, scale bar, and north arrow to ensure that these elements do not obscure key data.

The authors should also adjust the font size in various labels to ensure they are large enough to read. Even when the figure spans the entire page width, some labels are very small (e.g., RGT #'s, "spacecraft travel direction", the "10 km" scale bar, Surface elevation color bar, etc).

We will increase the font sizes as suggested.

Figure 1 (b)  What is R_5?

This is the location of the centroid of the fresnel zone of the reflected GNSS signal at a GNSS satellite elevation angle of 5*. We will increase the font and clarify this.

Figure 1 (c). This is a great cartoon of the kinematic GPS surveys. A small note is that H_R is capitalized in the figure but H_r is mentioned in the caption, update whichever to make sure symbology is consistent, also enlarge Z_track (next to the arrow) in the figure.

We will fix this.

L85-90: Here only 24 hours of data are collected, you can expect the largest errors in positioning at the beginning and end of an observation period, and depending on the processing procedure, at day-breaks. Were longer periods of data collected and it was found that 12-hours before/after was optimal?

Our analysis was based on 24-hour continuous static GNSS sessions, which provide stable PPP static solutions. While shorter sessions (e.g., 3 hours) can show increased variability due to the convergence period, in our case the use of full 24-hour data spanning midnight does not degrade the solution or show edge effects, and allows for a fully-converged singular static estimate.

Figure 2: the x marking the median is difficult to see, I suggest changing the symbology, perhaps a - that is longer than the underlying point measurements are wide would be easier to see?

We will make this larger.

Section 2.2 and 2.3: What are the CSRS reported horizontal and vertical (Zppp) errors? These should be included in these sections for both types of GNSS stations. I know this is discussed later on but is important to include here as well. You can refer readers to Section 3.3 for a more detailed discussion.

We will add text that explains this. It is thought that the reported errors overestimate the precision so we will refer the reader to 3.3 for more details.

L147-149: Here the authors state that observations from the full 24-hour period are used to determine this 1.4 cm measurement precision, my question is, are edge effects (at the beginning and start of your time series) are removed or special filtering is applied, etc? If you instead take a centered, say 12-hour period, do you get the same 1.4cm precision?

Because PPP processing is performed in continuous static mode, the fact that some sessions span midnight does not introduce edge effects. We have not experimented with longer data windows due to battery constraints, but we do observe a decrease in precision if the window is, say, only 3 hours instead of 24. This is the tradeoff between precision and the potential for the surface to change during the observation period. In general we opt for the longer (24 hr) observation period to increase precision.

Figure 3: Suggestion: In the caption indicate that the subplots are arranged by OGRE location from west to east. Adding a bold title or something similar to the 879* stations to indicate they are the stations along kinematic surveys would also be helpful for the reader.

We will update the caption to reflect this.

Figure 4 caption: typo: "for clarify" -> "to clarify?"

Fixed.

Can you put a point that matches the line color to mark these monthly observations? I agree the line is good for visual continuity but the points The up and downward pointing triangles are very hard to see. It appears they are centered on the line? Maybe offsetting these triangles either above or below all stations would make them more visible? Also maybe change the colors of some symbols specifically the x's marking Spot 3 and 4 which are difficult to see, particular the grey x or where there are overlaps. If the station colors are changed to a more muted color palette the symbols may be more easily seen? Regarding the "detected blowing snow" in particular, if present, blowing snow should be occurring across the entire study area and not necessarily concentrated on a few stations (due to high windspeed and abundance of snow). The presence of blowing snow could therefore be indicated at the top or bottom of the graph at each time period (by a symbol or shading vertically at that tilmestep) which would reduce some visual clutter.

We will standardize the symbology between the last two figures so that Spot 3 and 4 are consistent between both, and they are demarcated clearly.

L204: do you mean "Moreover"?

Corrected.

L218: "would" between "but" and "also"

Fixed.

---

## Author Response (AR2)

1. Is there any risk that the stake farm, which is proposed to substitute for the kinematic surveys, would interfere with the ICESat-2 data itself? Does the ICESat-2 footprint intersect any antennae exactly, and if so, does the ~20 cm antenna raised ~1 m above the snow register in the ICESat-2 photon cloud?

This GNSS station was placed away from the ICESat-2 overpass tracks, and used solely to assess footprint and biases. Thus, it does not pose interference here. I have added some brief phrases in both the main text and in the appendix to indicate that this OGRE and study site is not a part of the ICESat-2 comparison, and therefore does not affect ICESat-2 data.

As for the remaining stations, these all straddle the flight lines (Figure 1B) and therefore do not pose a problem either. As a thought experiment, I do believe an antenna in the flight track could register in the raw photon cloud, but be subsequently filtered out in higher level products, such as that which we use (ATL06). I have added a reference to Figure 1B in the text to direct the reader to that diagram, which highlights the geometry and layout of the GNSS station relative to the ground tracks of ICESat-2.

2. Please ascertain that the meanings of bimonthly (once every ~8 weeks) and semimonthly (once every ~2 weeks) are consistent with what you intend to communicate.

We log data semimonthly (2x per month) and monthly (1x per month). All instances of bimonthly are now replaced and corrected.

**Reviewer#1**
**We thank the reviewer for these valuable comments which will make this manuscript stronger. Please see our responses to each comment below in blue.**

The manuscript entitled "ICESat-2 surface elevation assessment with kinematic GPS and static GNSS near the ice divide in Greenland" by Pickell and others details a new use for the Open GNSS Research Equipment (OGRE) GNSS-IR stations—validation of ICESat-2 surface height measurements. This study first uses data from repeat kinematic GPS surveys at Summit Station, Greenland to assess the stability of ICESat-2 surface height measurements, finding a <0.01 m bias and <0.06 m precision between kinematic GPS and ICESat-2 observations. The authors then present a new validation method using GNSS-IR interferometric reflectometry to measure surface elevation coincident with ICESat-2 passes over their study site near Summit Station. Ultimately finding a good agreement between GNSS-IR and kinematic GPS observations and a bias in ICESat-2 measurements of ~0.09 m. These results indicate GNSS-IR can be deployed instead of kinematic GPS transects which are significantly more resource intensive.

I found the manuscript to be very well written with a robust methodology and results section wherein the results were well supported by the presented data. The manuscript would benefit from an elaboration on some methodology (detailed below) and a reorganization and expansion of the discussion section. I have detailed these points below and include some additional minor comments. Overall this

work is appropriate for the Cryosphere and I would therefore recommend minor revisions before publication.

Major comments

(1) I would like some clarification on the bias corrections applied to the dataset. Section 3.3, Lines ~155. Here a bias of 2.1+/-2.9 cm is reported, citing figure A1. I have a few questions regarding figure A1, but with the main concern that this bias is not temporally consistent. Is the bias applied across the full time series as stated? It appears from Figure A1b that there is indeed a large bias in 2022, a moderate bias in 2023, and somewhat in early 2025. Why not apply a variable bias correction? Or not at all as it is not convincing that a bias correction of 2.1+/-2.9 cm would improve your observations. Regarding Figure A1: Which OGRE is this data from? Where was the stake relative to this OGRE and the study area? Also was there a camera that allowed such frequent stake readings?

Assessing the temporal bias of GNSS-IR is an important and ongoing area of research in the reflectometry community. Traditionally, snow depth reflectometry studies have only measured the height of the GNSS mast, without much consideration to the true footprint of the reflectometry technique. We attempt to address the footprint issue by deploying an OGRE at the "Summit Station" marker in Figure 1, centered within a grid of 121 snow stakes spaced 8 m apart [See details in Pickell, Hawley & LeWinter 2025]. While these stakes are each measured to the nearest half centimeter, seasonal biases may still exist due to human error, seasonal pitting or mounding at the base of the stakes, etc. Thus, we must be cautious about the attribution of bias, and especially time-varying bias, to the reflectometry technique. Per the editor's comments, we have added text to indicate that this setup is unique and separate from the GNSS stations that fall under the ICESat-2 ground tracks.

We strike a compromise here by applying the overall mean bias correction to the OGRE, but we have adjusted our language and descriptions to make the answers to your questions clear and to point out that the most important comparison between the altimetry and OGREs is the low variability between the ICESat-2 and OGRE measurements, which is independent of the OGRE bias correction. [See line 245 in revision]:

"Consequently, we correct for this bias for all OGRE surface elevation estimates, noting that seasonal differences in biases are difficult to assess and this correction may be specific to this particular instrument-antenna setup in the dry snow zone; however, the most important metric of comparison is the temporal variability between the ICESat-2 and OGRE estimates, which are not affected by this bias correction."

In other words, we can think of it this way. Any arbitrary bias can be applied to the OGRE estimates (as long as it isn't time-varying), and the variability will not change between the OGRE and the satellite altimetry. It's this variability that we want to be small, and a small bias is also reassuring. However, a temporarily-varying bias correction will affect both the variability and the bias, and thus requires high

confidence to make this correction. As described in the appendix, we do not have the confidence to do this.

(2) Figures supporting methodology (either added to manuscript or supplement) L75-78: What is the range of track depth Z_track values due to vehicular weight depressing the snow at the beginning and end of your survey and the mean value used? L78-79: How did the laser range-finder measurements compare with the mean value of Z_track measured? Were there any systematic trends in this value? E.g., increasing depression along the survey track? A figure showing the Ztrack observations and the laser range finder observations would be beneficial.

Typical track depth values range from 0 to several centimeters depending on the survey and season/snow conditions, and in general correlate with corresponding qualitative data that detail recent snowfall (deeper tracks) or cold, clear conditions (shallower tracks, e.g.). The methodology for accounting for these variations is described in line 80. While the range of track depth values may be indicative of surface characteristics, we believe these values themselves do not warrant an additional figure since they primarily serve as a corrective term applied to the GNSS antenna height to obtain ground elevation, rather than as an independent variable such as accumulation or snow hardness, which could be illuminating of the surface characteristics.

We have updated the language to clarify our methodology in the discussion:

"The manual track-depth measurements (typically two per survey) provide a sparse estimate of the mean track depression, while the laser range-finder provides nearly continuous measurements along the track that allows us to assess the Gaussian variability characteristics. [To investigate this, we examine the data from the downward-pointing survey sled laser over two traverses. The $1\sigma$ variability in track depths was $\sim$2.0 cm. Since only two manual measurements are typically used to estimate the mean track depth for correction, this introduces a potential bias in the GPS-derived surface elevations due to under-sampling, with an estimated standard error of ±1.41 cm based on the laser-derived variability.]"

Unfortunately, we don't have more laser data from other survey dates from within this study to construct a full statistical analysis across the entire study period, and in general must take the routine, manual measurements at the beginning and end of each survey to be representative of the survey-wide track depth. However, our results from the laser rangefinder surveys are encouraging in that there is no systematic drift in track depth (e.g., indicative of changing snow conditions) for those particular surveys, which confirms the assumption that snow conditions are largely uniform in this region (although with some degree of sastrugi noise). Furthermore, the laser rangefinder is not referenced to the antenna ground plane datum: thus, it can assess variability of the surface relative to its own datum, but we cannot overlay track depth measurements with the laser range finder at this time.

(3) I found that the discussion would benefit from some slight reorganization to make it easier to follow for the reader, in particular by separating the discussion of temporal and spatial sources of bias which

are slightly intertwined (mainly the details presented in the middle to the end of paragraphs). Overall, the discussion does have a good organization by moving from those errors to external factors (e.g., blowing snow/surface roughness) then (correlated errors). But again, the authors should be careful to group like-ideas together (e.g., a discussion of the sensing footprint on line 248-249: is this an uncorrelated error or should it be moved to the spatial or surface roughness paragraphs? If it is kept in its current position the similarity of the sources of errors in the data should be made more evident.

We have structured sections of the discussion to more clearly separate temporal and spatial sources of bias. Specifically, details related to the sensing footprint will be adjacent to the discussion of spatial or surface roughness–related errors, so that similar sources of error are grouped together to improve readability.

(4) The Manuscript would also benefit from an expanded discussion of the implications, next steps, or synthesis of the work presented here. For example, the introduction and abstract mention that kinematic surveys are much more labor and resource intensive than the OGRE station deployment, since these results demonstrate OGRES are a useful tool to assess ICESat-2 surface elevations, what are the authors recommendations moving forward? I would be great to hear their thoughts for how future campaigns aiming to assess airborne/space-borne surface elevation measurements should proceed. These topics are particularly relevant given NASA's Snow4Flow program.

We agree that an expanded discussion of the implications and next steps would strengthen the manuscript. In the revised discussion section, we have highlighted that OGRE deployments offer a lower-cost, logistically simpler alternative to kinematic GNSS surveys for evaluating ICESat-2 surface elevations. A new paragraph was added as well. We note that while kinematic surveys remain valuable in specific settings, OGREs provide an efficient means to establish ground control across larger spatial and temporal scales, while reducing logistics [see final paragraph of discussion]. For example, radar altimetry validation could benefit from reflectometry given the reflectometry footprint is more agreeable with space-based radar, e.g. Or, with regards to Snow4Flow or NASA's NISAR mission, we can leverage the movement of static stations (which nominally is a challenge for co-located measurements) to validate ice flow and elevation change together. See added text:

"These results highlight the complementary capabilities of kinematic traverses and static GNSS stations for evaluating ICESat-2 surface elevations. Kinematic surveys provide spatially extensive, high-accuracy reference data, but their intensive logistical requirements limit temporal coverage. Static GNSS stations, by contrast, operate autonomously at relatively low cost, and our comparisons show they achieve biases and variabilities comparable to the kinematic approach. In addition, static stations overcome key challenges of the kinematic method by eliminating the need for timed traverses or manual track depth measurements. Beyond ICESat-2, radar altimetry validation could benefit from these stations, as the interferometric reflectometry footprint is well aligned with spaceborne radar measurements. Moreover, because static stations also resolve horizontal velocity, they hold particular promise for supporting new missions such as NISAR."

Minor comments

Figure 1 (a): I would suggest the north arrow be positioned on the top of the figure as it seems out of place near the scale bar. The legend for ICESat Traverse Route is somewhat misleading

We have experimented with alternative placements of the legend, scale bar, and north arrow to ensure that these elements do not obscure key data, but found that the placement of the north arrow in the top of the figure does not improve the readability of the figure, and furthermore makes the figure feel more off-balanced given the other data, key, inset, and information near the top of the figure.

The authors should also adjust the font size in various labels to ensure they are large enough to read. Even when the figure spans the entire page width, some labels are very small (e.g., RGT #'s, "spacecraft travel direction", the "10 km" scale bar, Surface elevation color bar, etc).

We have increased the font sizes as much as we think is possible, in response to this comment. We are open to exploring ways to make the font even larger if legibility becomes an issue on the proofs.

Figure 1 (b)  What is R_5?

This is the location of the centroid of the fresnel zone of the reflected GNSS signal at a GNSS satellite elevation angle of 5*. We have increased the font and added a sentence to highlight the significance of 5 degrees near line 100.

Figure 1 (c). This is a great cartoon of the kinematic GPS surveys. A small note is that H_R is capitalized in the figure but H_r is mentioned in the caption, update whichever to make sure symbology is consistent, also enlarge Z_track (next to the arrow) in the figure.

Fixed.

L85-90: Here only 24 hours of data are collected, you can expect the largest errors in positioning at the beginning and end of an observation period, and depending on the processing procedure, at day-breaks. Were longer periods of data collected and it was found that 12-hours before/after was optimal?

Unfortunately the OGREs were programmed to only collect data for 24 hours during each overpass period, based on the commonly adopted standard 24-hour continuous static GNSS session lengths used in PPP processing for optimal convergence time [text around line 90]. While shorter sessions (e.g., 3 hours) can show increased variability due to the reduced convergence period [line 95]. We also added text here to indicate that day-break effects and edge effects (which influence the convergence time) are important: "In this dataset, the reported 95\% ($\sim2\sigma$) vertical uncertainties are typically 0.7 to 0.8 cm, with higher uncertainties related to shorter convergence periods."

Figure 2: the x marking the median is difficult to see, I suggest changing the symbology, perhaps a - that is longer than the underlying point measurements are wide would be easier to see?

We have made this larger.

Section 2.2 and 2.3: What are the CSRS reported horizontal and vertical (Zppp) errors? These should be included in these sections for both types of GNSS stations. I know this is discussed later on but is important to include here as well. You can refer readers to Section 3.3 for a more detailed discussion.

Please see lines 150-151. ["While PPP solutions report uncertainties at mm-level, past studies in the cryosphere confirm that this is often an underestim ate (Khan et al., 2008)"] We have added additional text in 2.2 and 2.3 to point readers to 3.3 for more details.

Line 139: "Instead of using PPP estimates of uncertainty, we incorporate a more robust estimate of uncertainty described in Section 3.1."

And ~Line 109: "Thus, uncertainties stem from the reflectometry and PPP, which are addressed in Section 3.3."

Finally, line 155 includes vertical errors reported by CSRS. We do not include horizontal errors because we are only interested in the vertical positioning in this study.

L147-149: Here the authors state that observations from the full 24-hour period are used to determine this 1.4 cm measurement precision, my question is, are edge effects (at the beginning and start of your time series) are removed or special filtering is applied, etc? If you instead take a centered, say 12-hour period, do you get the same 1.4cm precision?

According to CSRS-PPP documentation about reduced errors due to midnight crossings, this crossing error is reduced. We have not experimented with longer data windows due to battery constraints, but we do observe a decrease in precision if the window is, say, only 3 hours instead of 24. This is the tradeoff between precision and the potential for the surface to change during the observation period. In general we opt for the longer (24 hr) observation period to increase precision [line ~90]: "In this dataset, the reported 95\% ($\sim$2$\sigma$) vertical uncertainties are typically 0.7 to 0.8 cm, with higher uncertainties related to shorter convergence periods."

Figure 3: Suggestion: In the caption indicate that the subplots are arranged by OGRE location from west to east. Adding a bold title or something similar to the 879* stations to indicate they are the stations along kinematic surveys would also be helpful for the reader.

We have updated the caption to reflect this.

Figure 4 caption: typo: "for clarify" -> "to clarify?"

Fixed.

Can you put a point that matches the line color to mark these monthly observations? I agree the line is good for visual continuity but the points The up and downward pointing triangles are very hard to see. It appears they are centered on the line? Maybe offsetting these triangles either above or below all stations would make them more visible? Also maybe change the colors of some symbols specifically the x's marking Spot 3 and 4 which are difficult to see, particular the grey x or where there are overlaps. If the station colors are changed to a more muted color palette the symbols may be more easily seen? Regarding the "detected blowing snow" in particular, if present, blowing snow should be occurring across the entire study area and not necessarily concentrated on a few stations (due to high windspeed and abundance of snow). The presence of blowing snow could therefore be indicated at the top or bottom of the graph at each time period (by a symbol or shading vertically at that tilmestep) which would reduce some visual clutter.

We have standardized the symbology between the last two figures so that Spot 3 and 4 are consistent between both, and they are demarcated clearly. Now, symbology to indicate beam number is consistent between the two figures.

L204: do you mean "Moreover"?

Corrected.

L218: "would" between "but" and "also"

Fixed.

**Reviewer #2**
**We thank the reviewer for these valuable comments that improve this manuscript. Please see our responses to each comment below in blue.**

General comments

This is a great study. We need regular altimetry validation studies that use observations from kinematic GNSS, and this study which uses a unique dataset that samples surface height change in winter and in summer fills an observational gap.  I think the manuscript should be accepted with minor revisions. I also have mostly general comments that the authors can take or leave.

My main criticism is that I think the manuscript could go into more detail about the surface elevation trend that you observe at Summit. Do we see an increase or decrease in mean surface height over the 2-year period you have observations? The density of your observations in space and time provide a

unique constraint that I think could be used to describe in more detail the subtle surface elevation change signals that are propagating into the interior from more well documented change from the margins. Could you add in the conclusion or the discussion a short subsection that describes these elevation changes and speaks to the significance of this observation in the context of Greenland/Summit accumulation derived from reanalysis and your observations? Consider adding a section in the discussion:

Interior surface-height change detected by the OGRE network

Beyond bias characterization, the OGRE time series document a net rise in the interior surface of the Greenland ice sheet during 2022-2024..”

We grappled with whether to include information about elevation change in this manuscript as studying not only absolute biases but also the ability of ICESat-2 to correctly capture elevation trends through time is important. Furthermore, understanding the processes that contribute to elevation change in interior Greenland are important for many glaciological and climate applications.

In fact, we have a manuscript in preparation that examines these points on elevation change. We feel this belongs in a separate manuscript because the processes that contribute to elevation change are complex. We must take into account firn thickening/thinning, ice thickening/thinning, in addition to surface processes such as accumulation, thus requiring multiple additional datasets and analyses that we feel would distract from the scope of this manuscript.

It might be outside the scope of this study, but in the future, could you consider using dual frequency GNSS from ground-based radar surveys and UNAVCO kinematic GNSS data to increase the number of GNSS/IS2 crossover points? Much of this data exists and is fully processed on UNVACO/CReSIS servers already, and though most of this data was collected without monitoring sinking from sleds, the sled design/geometry is well constrained and photos from different seasons could be used  to calculate the sinking term and augment the year-round surveying described here for summer months over different regions of Antarctica and Greenland for environments where the surface conditions are more rough near the coast.

This is a good suggestion: we have examined several existing on-ice GNSS studies, including the SMM3 UNAVCO/Earthscope station at Summit and the GLISN stations in south Greenland. These static stations can use reflectometry to derive the surface elevation. However, they were not deployed underneath ICESat-2 overpass lines and therefore required additional corrections for slope that make them too imprecise for this study. These sites, in addition to other kinematic surveys, may be very suitable for radar altimetry comparisons, given the larger footprint and potential to fall underneath the ground tracks of these other satellites.

One other question I have is connected to the methodology and processing of the kinematic GNSS data and -IR data. I think with a base station at summit and the network in Greenland maintained by UNAVCO,

it should be possible to process the kinematic GNSS using TRACK relative to base station solutions using software like GAMIT/GLOBK (or public solutions from repositories that host GNET and summit data). In the case of the OGRE, processing with GAMIT/GLOBK as part of a larger solution for Greenland may improve the relative surface height estimates and could be worth considering in the future. If you need help setting this up for future studies, we can connect after the review period (I don't think it affects any of your main conclusions here).

Any suggestions to make the data more useful in future studies are appreciated and we would happily explore this suggestion further.

The only other delicate suggestion I have is to perhaps make less strong claims about the originality of the autonomy of these systems. For instance, someone likely from unavco or pascal is raising these sensors to make it possible to do GNSS-IR over multiple seasons. This is a lot of work, and it's been done for quite a while at Summit, but also more remote sites. For instance, take the second paragraph of the conclusion:

"We also present an autonomous method of retrieving ground-based surface elevation estimates using GNSS interferometric reflectometry with a standard GNSS receiver, mounted on a mast in the snow."

This language makes it seem as though this is the first use of GNSS-IR for monitoring surface height change of ice sheets when most of the methods you've described are well established (and I think still require people to service the instruments?). I think these sentences could be modified to emphasize the novel application (surface tracking/altimetry validation) using an established method. I don't think This was intentional, and my suggestion is just to make this more clear.

We appreciate this suggestion. Indeed, several papers have made use of UNAVCO on-ice infrastructure for reflectometry (e.g., Larson, M MacFerrin, T Nylen 2020), who first suggest the applicability of their methods to altimetry validation. We have adjusted our language accordingly in the introduction [line 45-additional citations added] and elsewhere (change "present" to softer word – "use" in line 285) while maintaining the novel aspect here: that these stations were designed and deployed specifically for this purpose (e.g., geolocated along ICESat-2 paths with appropriate antenna heights). We also changed a word in the abstract: "introduce" -> "describe". We welcome feedback on if this strikes the correct tone; again we want to highlight both the novelty of this study and the existence of GNSS-IR as an established technique elsewhere in the cryosphere.

Below are minor suggested changes for style and content:

Figure 1: In panel B, it appears most of the reflections are coming from within this azimuth angle of 5 degrees, but that this zone doesn't overlap in this case with the icesat2 passes. Could similar figures be made for all sites to show how where the measurements you're making are relative to the the icesat-2 tracks.

Given the relative compactness of our network (~30 km east-west), the reflection zones and azimuth angles are mostly consistent from station to station. The fact that the elevation angle window emphasizes the surface area closer to the instrument than the ICESat-2 paths is perhaps our largest source of uncertainty with this method [see discussion] and we have adjusted our language accordingly: see line 170-175 and re-worked the discussion (235-... "unique to GNSS-IR method, the sensing footprint shrinks as the pole becomes buried, potentially increasing the sensitivity to localized topography or surface roughness, thereby influencing the stability and representativeness of the derived elevation"

And Caption (B) is changed to indicate that this is the same ('typical') setup for each site. Hopefully this indicates that each station is set up in the same way: in the middle between the two beams.

LN: 7-11 consider removing autonomy, and defining GNSS, GNSS-IR. Also choose GNSS or GPS (as I think you probably use solutions from all the satellites not just GPS?)

"while reliable, these surveys are resource-intensive. We introduce an alternative, novel validation method using Global Navigation Satellite System (GNSS) Interferometric Reflectometry…"

GPS refers to the kinematic survey with the R7 receiver, which is old enough to only track GPS/GLONASS and here only tracks GPS: we define GPS when first mentioned in the introduction (line 30) and GNSS when first mentioned in the introduction (line 37). We will consult with the editor about acronym definition here, in the abstract given that these are relatively well known terms.

L12-16 Consider quoting the bias and standard deviation directly. Revealing mean bias of *** +/- ** cm relative to ICESat-2.

These quantified details for each methodology/technique may be too detailed for these initial sentences given the diversity of techniques.

L23-31: "By contributing a complementary geographic setting to antarctica.."

This is a long and awkward sentence to me -consider revising, and reframing around Summit as a legacy validation site and link to icebridge and icesat-1.

Fixed.

LN 112-118 (Section 3.1): "The median standard deviation of each pseudo-static point..

This was a sentence that I felt could be shortened and combined with the second sentence:

The median standard deviation of the pseudo-static points was 0.8 cm (n=****), consistent with prior Summit estimates (0.9 -1.8 cm).

We have considered this suggestion, but feel that too much information was lost (e.g., what median standard deviation actually means, the difference in receivers Trimble). So, the outcome is the combined sentences read run-on if we do not drop this information.

LN 158- 166 (Section 3.4): "We follow the same method described above to compare ICESAT-2…"

Consider changing for clarity to: For ATL06 – ORGE, we apply the same filters (quality.= 0 ; …, but we use a 60 m search radius (beam-pair spacing 45 m) and a linear cross-track interpolation between Spots 3-4 at the OGRE latitude.

Changed.

LN 233- 239 (Section 5): Consider changing "When we segregate…" to "Separating overpasses flagged for blowing snow or clouds does not affect bias or precision."

Changed.

Appendix A: It looks like there was an idea that was not finished or completed. What was the example the authors intended to include here (e.g. …).

Fixed.

Minor comment (not necessarily for a single section):

Can you include a summary table of the parameters you used in the GNSSrefl code. It would also be great to include figures of the Fresnel zone for each receiver as this shows really explicitly the area that you sample from. Height solutions can be sensitive to fesnel zones and the threshold azimuth angle, and recording all this information in a table could help users who want to replicate this kind of study quickly (a lot of this is already well documented in the code). Also include information about which frequencies (likely both?) and constellations were used in the reflection solution.

We have ensured that the processing parameters are properly detailed for easy replication: see lines 98-100 and 101 to 102, added to address this. I can change this to table form, but I found these parameters are succinct enough to be addressed in sentence form. The Fresnel zone is highlighted in Figure 1b.

**Copy edits:**

LN 29: bi-monthly -> bimonthly, and consider rewriting for clarity

Changed to semimonthly.

LN 44: Consider citing Larson & Nievinski (2013), Seigfried et al., (2017), Hoffman et al., (2025), Trine et al., (2024), which have used -IR to measure accumulation. GNSS-IR is a powerful measurement technology that is still underused in the glaciological community. Citing these other studies can bring awareness to this method and how it can be used in validation studies of surface height change and to understand near surface accumulation and firn densification.

Added these citations. We could not find a reference for Trine 2024 but included Dahl-Jensen (Trine) 2010.

LN 190: delete temporal

Done.

Figure 4 caption: Clarify -> clarity

Done.

Section 5: snow pack -> snowpack.

Done.

Throughout: Use ~ throughout to approximate value.

Done.

Throughout: Overflight -> overpass .. I'm not sure what the community standard is here. Flight seemed odd to me, but I could be wrong.

Standardized to overpass.

Throughout: Sub daily -> subdaily

Done.

Throughout: Include space before units.

Done.

Throughout: 1-\sigma -> 1\sigma

Done.

Throughout: Consider abbreviating y^-1 to yr throughout and being consistent with abbreviation of s y, and day (d). I defer to the editor on this. I'm not sure what the best practices are for the cryosphere.

We will change to yr and double check with the editor.